# Return to Work, Fatigue and Cancer Rehabilitation after Curative Radiotherapy and Radiochemotherapy for Pelvic Gynecologic Cancer

**DOI:** 10.3390/cancers14092330

**Published:** 2022-05-08

**Authors:** Eva Meixner, Elisabetta Sandrini, Line Hoeltgen, Tanja Eichkorn, Philipp Hoegen, Laila König, Nathalie Arians, Jonathan W. Lischalk, Markus Wallwiener, Ilse Weis, Daniela Roob, Jürgen Debus, Juliane Hörner-Rieber

**Affiliations:** 1Department of Radiation Oncology, Heidelberg University Hospital, 69120 Heidelberg, Germany; Elisabetta.Sandrini@med.uni-heidelberg.de (E.S.); Line.Hoeltgen@med.uni-heidelberg.de (L.H.); Tanja.Eichkorn@med.uni.heidelberg.de (T.E.); Philipp.Hoegen@med.uni-heidelberg.de (P.H.); Laila.Koenig@med.uni-heidelberg.de (L.K.); Nathalie.Arians@med.uni-heidelberg.de (N.A.); Juergen.Debus@med.uni-heidelberg.de (J.D.); Juliane.Hoerner-Rieber@med.uni-heidelberg.de (J.H.-R.); 2Heidelberg Institute of Radiation Oncology (HIRO), 69120 Heidelberg, Germany; 3National Center for Tumor Diseases (NCT), 69120 Heidelberg, Germany; 4Department of Radiation Oncology, Perlmutter Cancer Center, New York University Langone Health, New York, NY 10023, USA; Jonathan.Lischalk@nyulangone.org; 5Department of Gynecology and Obstetrics, Heidelberg University Hospital, 69120 Heidelberg, Germany; Markus.Wallwiener@med.uni-heidelberg.de; 6Social Service Counselling, University Hospital Heidelberg, 69120 Heidelberg, Germany; Ilse.Weis@med.uni-heidelberg.de (I.W.); Daniela.Roob@med.uni-heidelberg.de (D.R.); 7Heidelberg Ion Therapy Center (HIT), 69120 Heidelberg, Germany; 8German Cancer Research Center (DKFZ), Clinical Cooperation Unit Radiation Oncology, 69120 Heidelberg, Germany

**Keywords:** endometrial cancer, definitive radiotherapy, adjuvant chemoradiation, toxicity

## Abstract

**Simple Summary:**

Rehabilitation from cancer treatment and therapy-induced toxicity requires individualized and specialized expertise. Beyond the resolution of treatment-related morbidity, socio-economic and psychological factors must be considered, and lifestyle or household characteristics can have a notable impact on the gradual return to normality and return-to-work rates following cancer therapy. A better identification of patients at a higher risk of prolonged impairment, and a more in-depth understanding of the impacts of treatment is needed to optimize post-therapy recovery. We aim to add to a limited body of literature exploring the posttreatment rehabilitative factors for women following curative radiotherapy for primary gynecologic malignancies. Herein we observed therapy-induced pain and fatigue were significantly more likely to interfere with return-to-work rates. Social support services and post-treatment inpatient cancer rehabilitation programs were helpful in keeping patients connected to their professional lives. Mental issues and the development of depression during follow-up remains an issue particularly for younger patients.

**Abstract:**

Pain, fatigue, and depression are a common cluster of symptoms among cancer patients that impair quality of life and daily activities. We aimed to evaluate the burden of cancer rehabilitation and return-to-work (RTW) rates. Tumor characteristics, lifestyle and household details, treatment data, the use of in-house social services and post-treatment inpatient rehabilitation, and RTW were assessed for 424 women, diagnosed with cervical, uterine, or vaginal/vulvar cancer, receiving curative radio(chemo)therapy. Progression-free RTW rate at 3 months was 32.3%, and increased to 58.1% and 63.2% at 12 and 18 months, respectively. Patients with advanced FIGO stages and intensified treatments significantly suffered more from acute pain and fatigue. A higher Charlson-Comorbidity-Index reliably predicted patients associated with a higher risk of acute fatigue during RT. Aside from the presence of children, no other household or lifestyle factor was correlated with increased fatigue rates. Women aged ≤ 45 years had a significantly higher risk of developing depression requiring treatment during follow-up. Post-treatment inpatient cancer rehabilitation, including exercise and nutrition counseling, significantly relieved fatigue symptoms. The burdens for recovery from cancer therapy remain multi-factorial. Special focus needs to be placed on identifying high-risk groups experiencing fatigue or pain. Specialized post-treatment inpatient cancer rehabilitation can improve RTW rates.

## 1. Introduction

Pelvic gynecologic malignancies comprise a significant and increasing number of cancers in women, with about 604,000 new cases of cervical, 417,000 new cases of uterine, 45,000 new cases of vulvar, and 17,900 new cases of vaginal cancer diagnosed in 2020 [1,2]. Whereas cancer prevention, detection, and treatment represent a large proportion of current oncologic research, there is little evidence regarding the integration of social services and rehabilitation for cancer survivors.

Treatment concepts depend on the International Federation of Gynecology and Obstetrics (FIGO) classification stages, and can include surgery and definitive or postoperative radiotherapy (RT), systemic therapies, and chemoradiation. For primary curative treatment, surgery with postoperative RT or definitive radiochemotherapy often provide the cornerstone of curative therapy. Pelvic RT treatment-related side effects have been mitigated through the use of modern intensity-modulated techniques, but these can impair the quality of life nonetheless and result in late genitourinary and gastrointestinal side effects [3,4]. Furthermore, fatigue is a common symptom in cancer treatment, particularly in younger patients with advanced tumors and can interfere with normal activities of daily living including the ability to work [5,6]. The diagnosis of cancer is known to be associated with a psychosocial burden, and patient and treatment characteristics can have a significant impact, not only on access to and utilization of self-care and social services, but also on rehabilitation [7].

Long-term cancer survivors have been shown to experience work changes during follow-up, which can be accompanied by financial or social impairments [8]. Furthermore, socio-economic factors, such as household characteristics, health status, tumor stage, and adjuvant oncologic therapies, have been shown to correlate with return to work (RTW) rates, but data are inconsistent [9,10,11]. Return to work in survivors following cancer treatment has been reported with mean overall rates of 62%, and significantly depend on the primary tumor site [12]. For example, head and neck cancer was seen as a risk factor, while men with testicular cancer had fewer treatment-related burdens [12]. Following primary surgery, 59% of breast cancer patients had returned to work after 10 months; in another series, 35% of women with early-stage breast cancer did not until after one year [11,13]. While physical impairments including the movement restrictions of the arms after breast cancer therapy impaired RTW, this could not be determined for the presence of fatigue [13]. Moreover, data are mostly validated for breast cancer, in pelvic gynecologic cancers, detailed analyses of the aforementioned quality of life factors is lacking [9,11,14]. The aim of the current study was to analyze high-risk groups in need of more specialized care and to assess factors associated with impaired cancer rehabilitation, as well as barriers for return to work, focusing on women with curative radio(chemo)therapy for primary pelvic gynecologic malignancies.

## 2. Materials and Methods

### 2.1. Patient and Treatment Characteristics

We retrospectively assessed women who were treated with curative RT or chemoradiation at a single institution between January 2017 and December 2021. Our analysis was approved by the local ethics committee (S-453/2021, approval date: 7 June 2021). Only women diagnosed with primary pelvic gynecologic malignancies, including cervical, uterine, vulvar, and vaginal cancer were included. Palliative intent treatment and patients diagnosed with metastatic disease were excluded. At diagnosis, patients were staged according to the corresponding International Federation of Gynecology and Obstetrics (FIGO) classification and the TNM American Joint Committee on Cancer staging system; management decisions were determined by a multidisciplinary tumor conference.

Patient characteristics, comorbidities, demographics, and socio-economic factors and lifestyle factors, were reviewed in detail. The smoking status, educational level, household details, marital status, and number of children were assessed in routine clinical or during in-house social service consultations.

For baseline measures, current working status, practiced profession and salaried- or self-employment characteristics at the time of first diagnosis were routinely reviewed at the first outpatient appointment at our radio-oncologic department or retrospectively assessed during in-house social service consultations or during follow-up appointments. Return to work status was evaluated during treatment and follow-up by reviewing in-house or referring physician’s notes, inpatient cancer rehabilitation discharge letters and social workers’ databases and documented as successful at the time point of the first vocational reintegration.

The Charlson Comorbidity Index (CCI) was calculated for each patient for baseline measures as defined prior to RT [15]. With respect to CCI, an adjusted scoring for the category of “solid tumor” was utilized; points were counted only if a second malignancy, other than the treated one, was present.

Body weight was assessed at radiation treatment start and end as well as at the first follow-up appointment. Furthermore, the presence and extent of weight loss up to three months before the start of RT was collected. Each individual’s body mass index was calculated as follows: BMI (kg/m^2^) = weight/height × height. Finally, clinical and pathological tumor data and treatment characteristics were reviewed.

### 2.2. Specialized In-House Social Service and Inpatient Cancer Rehabilitation

Specialized in-house social service consultation was offered routinely to each patient during treatment to identify and provide knowledge and professional interdisciplinary assistance for physical, psychological, administrative, social, and financial burdens. Further, to simplify the access to rehabilitation services and legal claims, a team of social workers helped to claim the costs of the inpatient cancer rehabilitation programs and necessary medical equipment to be covered by the patient’s health or pensions insurance company according to local payers’ policies and governmental regulations.

Post-treatment inpatient cancer rehabilitation was performed by physicians and physical therapists at specialized oncologic care center, which were either patient’s preferred or provider’s recommended facilities. Prior to admission, insurance providers had checked the physician’s confirmation of necessary pre-conditions for a patient’s ability to attend and tolerate intensive physical rehabilitation programs. Each facility addressed concepts of physical exercise as well as medical and nutrition education during an inpatient stay to maximize physical and social functioning according to each individual’s needs and appropriate goals.

### 2.3. Radiotherapy

Radiotherapy was delivered using external beam radiotherapy (EBRT) with 6–18 MV photons, using intensity-modulated RT (IMRT) in an adjuvant or definite curative fashion in once-daily fractions. Chemotherapy was given intravenously, simultaneously or sequentially, according to multidisciplinary tumor conference recommendations. For selected patients, high-dose rate brachytherapy was used alone or in combination with EBRT, according to each patient’s individual risk factors. Radiation treatment planning was based on computer tomography (CT) imaging along with magnetic resonance imaging (MRI), if available. For macroscopic tumor masses, a simultaneously integrated or sequential boost was allowed.

### 2.4. Toxicity and Oncologic Outcome

The evaluation of oncologic outcomes during follow-up included the review of clinical data, imaging, and referring physician notes. Recommended follow-up intervals were every three months during the first and second year after treatment, every 6 months in the third year and once a year after that for at least five years. Follow-up times until last contact were assessed and reported for the overall group and additionally for the working subgroup. For the evaluation of toxicity and symptoms, baseline was defined prior to the start of RT as assessed in clinical routine at the first outpatient appointment at our radio-oncologic department. The presence of depressive symptoms was documented by physicians during routine visits at baseline and follow-ups and included the requirements of medical or psychological treatment, while the presence, occurrence and extent of fatigue symptoms were graded according to Common Terminology Criteria for Adverse Events (CTCAE, version 5.0). Acute (up to 90 days after the start of RT) and late (>90 days) toxicities were graded according to the CTCAE during RT and follow-up and included gastrointestinal disorders, pain, nausea/vomiting, urinary toxicities as well as dermatitis/mucositis, lymphedema, and fatigue. Progression-free survival was assessed and defined using the time from the start of RT until any local or systemic progression.

### 2.5. Statistical Analysis

Weight loss prior to RT and during RT, smoking status, higher educational level, marital status, the presence of children, and depression requiring treatment were considered categorial data (yes vs. no). The Charlson Comorbidity Index was considered categorial data (≤5 vs. >5), while the number of fractions and RT doses were considered continuous variables. Acute toxicity was presented as dichotomous (present vs. not) but also graded according to CTCAE grading as stated in the corresponding results. The normal distribution of the variables was analyzed. Quantitative data and group characteristics without a normal distribution were compared using the Mann–Whitney U tests for continuous data and results presented with median and quartiles. The Pearson Chi-squared (Χ^2^) tests was used for categorical data. The log-rank test and uni- and multivariate Cox regression with hazard ratios (HR) were used with a 95% confidence interval (95%CI). Kaplan-Meier (one minus) estimates were assessed for RTW rates, women who did not re-start work before the last contact were censored. Receiver operating characteristic (ROC) curve analysis was performed to calculate cut-off age values. Age was dichotomized (<45 vs. ≥45 years) for further statistical analyses. A *p*-value less than 0.05 was being considered statistically significant. For statistical calculations, IBM’s statistical software, SPSS, was used (version 28, Armonk, NY, USA).

## 3. Results

### 3.1. Patient and Treatment Characteristics

A total of 424 women with a median age of 61 years receiving curative radio(chemo)therapy, between January 2017 and December 2021, were identified. Median follow-up time was 11.6 (range: 0.2–65.6) months. Endometrial cancer was the most common histology (*n* = 205, 48.3%), followed by cervical cancer (*n* = 166, 39.2%), and vaginal or vulvar cancer (*n* = 53, 12.5%). Treatment was delivered as RT only for 195 (46.0%) women, and simultaneous or sequential radiochemotherapy for 181 (42.7%) and 48 (11.3%) women, respectively. A total of 269 (63.4%) women received upfront surgery and 155 (36.6%) were treated with a definitive RT concept. Radiotherapy was applied in a median of 28 (range: 1–38) fractions, with a median dose of 45 (6–60) Gy and a median boost dose of 20 (9–63) Gy. Inpatient treatment was necessary in two-thirds of patients (*n* = 137, 66.5%). Three patients did not complete treatment as planned due to treatment-related side effects and patients’ refusal for treatment completion. Eighty-two (19.3%) patients experienced local or systemic failure during follow-up. Patient and treatment details are listed in Table 1.

Household details revealed that 81.7% of patients were married or had a partner, 7.1% of the women were living alone, and 86.5% had children with a median number of one (range: 0–6) child. Active nicotine abuse was documented in 59 (13.9%) women, whereas 23 patients (5.4%) had stopped smoking prior to RT. At the time of diagnosis, 33 patients (7.8%) suffered from depression. In this case, 16 women (3.8%) developed new depressive symptoms during follow-up that required medical or psychological treatment. Women aged less than 45 years (*n* = 48) at the start of RT were significantly more in danger of developing depression (*p* = 0.001) after the end of treatment than older women. Detailed statistical results are listed in the Supplementary Material in Appendix A.

The population was predominantly White/Caucasian (*n* = 401, 94.6 %) and only minimal representation of North Asians (*n* = 13, 3.1%), South Asians (*n* = 3, 0.7%), Sub-Saharan (*n* = 5, 1.2%), and other (*n* = 2, 0.4%). Thus, subgroup analyses were not considered to be reliable and were not performed for ethnicity groups.

### 3.2. Treatment-Induced Pelvic Toxicity

The most common acute RT-induced toxicities were low grade (grade 1 + 2) gastrointestinal disorders (*n* = 147, 34.7%), pain (*n* = 121, 28.5%), nausea/vomiting (*n* = 102, 24.1%), urinary toxicity (*n* = 101, 23.8%), dermatitis/mucositis (*n* = 61, 14.4%), and lymphedema (*n* = 13, 3.1%). High grade 3 toxicity was relatively uncommon and present in 18 (4.2%) women with gastrointestinal disorders, 16 (3.8%) patients with dermatitis/mucositis, 9 (2.1%) women with urinary toxicity, 3 (0.7%) with pain, and 1 (0.2%) with lymph edema. No grade toxicity ≥ 4 was observed. Sixty-two (14.6%) patients gained weight during RT, while 103 (24.3%) had a lower body weight at the end of treatment, with a median weight loss during the course of RT treatment of 0 (range: −17.9 to +12.2) kilograms. Median change of Karnofsky performance score during RT was 0% (range: −30% to +20%).

Women with a higher Charlson Comorbidity Index score suffered more often from acute nausea (*p* = 0.001) and weight loss during RT (*p* = 0.005) and had higher rates of inpatient treatment necessity during RT (*p* < 0.001).

Nausea, gastrointestinal toxicity, and dermatitis/mucositis were more present in women with higher FIGO stages (nausea: *p* < 0.001; gastrointestinal: *p* < 0.001; dermatitis/mucositis: *p* < 0.001) and intensified treatments (chemotherapy (nausea: *p* < 0.001; gastrointestinal: *p* < 0.001,dermatitis/mucositis: *p* < 0.001), more fractions (nausea: *p* < 0.001, gastrointestinal: *p* < 0.001, dermatitis/mucositis: *p* < 0.001), and higher RT doses (nausea: *p* < 0.001, gastrointestinal: *p* < 0.001, dermatitis/mucositis: *p* < 0.001).

Furthermore, the presence of nausea and gastrointestinal disorders were more common in women that experienced weight loss before the start of RT (nausea: *p* = 0.042, gastrointestinal: *p* = 0.043) and were significantly more often accompanied by fatigue symptoms (nausea: *p* = 0.014; gastrointestinal: *p* = 0.014) and weight loss during RT (nausea: *p* < 0.001, gastrointestinal: *p* < 0.001).

Urinary toxicity was also significantly increased for the above-mentioned intensified treatments (chemotherapy (*p* = 0.024), more fractions (*p* < 0.001), and higher radiation doses (*p* < 0.001), but was also significantly more present among women aged less than 45 years (*p* = 0.036) and smokers (*p* = 0.029). The presence of lymphedema did not show a correlation to tumor or RT treatment factors, but was significantly higher in smokers (*p* = 0.011).

### 3.3. Fatigue and Pain

Prior to RT treatment, 27 (6.4%) women suffered from grade 1 and three (0.7%) women suffered from grade 2 fatigue. At the end of RT, 118 women (27.8%) had experienced grade 1, 35 (8.3%) patients experienced grade 2, and 1 woman (0.2%) experienced grade 3 fatigue. At the first follow-up, fatigue was present in 147 (34.7%) women at grade 1, 31 (7.3%) at grade 2 and 1 (0.2%) at grade 3. During the follow-up period, fatigue symptoms were relieved in 94 women after a median time of 2.3 months. Women aged over 45 years (*p* = 0.025) and childless women had less acute fatigue symptoms (*p* = 0.006) during RT, other household and lifestyle factors did not show significant correlations.

Charlson Comorbidity Index prior to RT was significantly higher in patients in which fatigue appeared during RT (0.047) and during follow-up (*p* = 0.028). Women that received higher RT doses (*p* < 0.001), more fractions (*p* < 0.001), and simultaneous systemic therapy (*p* = 0.015) were more likely to suffer from fatigue during RT. No significant correlation between fatigue and treatment characteristics were found for boost doses, simultaneously integrated boost application, or an extensive para-aortic planning volume.

Women with fatigue showed a higher, but not significant deterioration of Karnofsky performance score during RT (*p* = 0.061), suffered more often from weight loss (*p* = 0.026), and needed inpatient treatment (*p* < 0.001) more often. Patients that underwent specialized inpatient rehabilitation therapy after the end of RT achieved relief for their fatigue symptoms during follow-up significantly more often (*p* < 0.001). The time until relief of fatigue symptoms during follow-up was longer in women that had a weight loss of more than 5 kg before the start of RT (*p* = 0.045). The presence of higher FIGO stages (*p* = 0.006), definitive concepts (*p* < 0.001), intensified treatment with RT volumes, including para-aortic sites (*p* = 0.012), the application of chemotherapy (*p* < 0.001), more fractions (*p* < 0.001), and higher doses (*p* < 0.001) led to significantly more acute pain during treatment and higher numbers of inpatient treatments (*p* < 0.001). Patients that suffered from acute pain during RT had significantly more often weight loss during RT (*p* < 0.001). Age, household, and lifestyle factors did not reflect the extent of pain symptoms.

### 3.4. Social Services, Socio-Economic Factors and Inpatient Specialized Rehabilitation

Specialized in-house social service consultation was offered to all women during treatment and 56.8% (*n* = 241) of the women made use of this service after a median time of 27 days after the start of RT. The age of women who received social service counseling and those who rejected it, was not statistically different. Patients with acute nausea (*p* = 0.011) during RT and higher Charlson Comorbidity Index (*p* = 0.039) received in-house counselling more often.

Significant differences could not be found concerning the number of patients treated with RT per year, or for the numbers of patients with social service consultations per year in the observed period of 2017 to 2021.

With a median time of 40 days after the end of RT, 122 (28.8%) patients underwent specialized inpatient cancer rehabilitation for a median time of three weeks. Women registered significantly more often for this if they were younger than 60 years (*p* = 0.031), had a higher educational level (subgroup *n* = 245, *p* = 0.022), had acute genitourinary toxicity (*p* = 0.031) during RT, experienced longer RT treatment times with more fractions (*p* = 0.032), and after social in-house counselling had taken place (*p* < 0.0001).

### 3.5. Working Status and RTW Rates

At baseline, a total of 191 (45.0%) women were working full- or part-time, 188 (44.3%) patients were already retired and 45 (10.6%) patients were of working age, of which 38 (9.0%) were housewives and 7 (1.7%) were unemployed. Median follow-up for the working subgroup was 20.8 months (range: 1.2–62.4) months. 16 patients (8.3%) within this subgroup of working women were lost-to-follow-up after 15 months, for the remaining 176 women (91.7%) full follow-up of at least 18 months was assessable.

Within the active working group (*n* = 191), more detailed characteristics regarding salaried- or self-employment was assessable for 142 working patients: 133 (93.7%) were salaried and 9 (6.3%) were self-employed.

During follow-up period, 101 (52.9%) women returned to work after a median time of 3.2 (range: 0–33.1) months, while 91 (47.6%) patients did not return. The mismatch of 1 woman (191 working at baseline, 192 observed during follow-up) was caused by one woman that was working age and counted as unemployed at baseline assessment prior to RT (as she had lost her job only a few days prior to the start of definitive RT), but was able to return to employment only a few days after the end of RT.

Among the group of 91 patients, that did not return to work, 36 (39.6%) experienced local or systemic tumor progression, requiring cancer treatment. Progressive disease was significantly (*p* < 0.001) associated with lower RTW rates (Figure 1A). Table 2 lists the RTW rates at different timepoints for the overall group of working patients (Table 2a) and among the 155 progression-free women (Table 2b).

RTW rates were significantly lower in patients with a higher FIGO stage (*p* = 0.003) and definitive treatment concepts (*p* = 0.024). Moreover, RTW rates were significantly lower in patients who experienced fatigue during RT (*p* = 0.048) (Figure 1B) and during follow-up (*p* = 0.015), in patients with acute pain during RT (*p* = 0.033) and in patients with depression (*p* = 0.010) (Figure 1C). Significantly higher rates could be found in self-employed women (subgroup *n* = 142, *p* = 0.029) and for patients with a BMI within normal ranges at the end of RT (*p* = 0.014) (Figure 1D).

Return to work rates were higher in patients that experienced improvements in terms of fatigue symptoms during follow-up (*p* < 0.001) and that underwent specialized inpatient rehabilitation after RT (*p* < 0.001).

Household details, the presence of children, marital status, or age did not significantly influence re-entry into employment. Return to work rates did not vary significantly throughout the observed time, from 2017 to 2021.

While the Charlson Comorbidity Index was not able to reliably predict RTW (*p* = 0.336), the pre- (*p* < 0.001) and post-treatment (*p* = 0.015) Karnofsky performance scores were a significant measure for women with successful RTW.

## 4. Discussion

Our study found that rehabilitation in women with RT for pelvic gynecologic malignancies highly depends on the intensity of cancer treatment and especially the appearance of fatigue and pain; social service support, including inpatient specialized care, can help to improve rehabilitation outcomes.

While RTW rates for other tumor sites have been examined in the past, data for uterine, cervical, and vaginal/vulvar cancer are lacking. Our data show that only 58.1% of progression-free women returned to work at 12 months. This indicates a need for a better identification of patients at a higher risk of prolonged impairment, and a more in-depth understanding of the impacts of cancer treatment to optimize interventions for post-therapy recovery. Indeed, during the course of curative radiotherapy for pelvic malignancies there is a complex interplay of individual patient, socioeconomic, supportive, cancer, and treatment factors which all have manifestations and the patient’s ultimate outcome.

Only the presence of a normal BMI reliably reflected increased RTW rates in our analysis, while smokers experienced significantly more urinary and lymphedema toxicity in our population. Furthermore, childless women suffered less from fatigue. However, the association between socio-economic factors and RTW rates remains controversial throughout the published literature [9,11]. Nevertheless, advocating for concepts of education and rehabilitation, which target a healthy lifestyle, exercise, nutrition, and fitness can have a protective effect and positive impact on oncologic outcomes and side effects. Furthermore, with respect to physical and mental functions, unmodifiable factors including age and comorbidities require specific focus and targeted management independent of the tumor diagnosis itself.

Historical literature has explored rehabilitation and return to work outcomes in the oncologic sphere, however, there is a lack of data surrounding gynecological malignancies. Certainly, the anatomical disease site dictates acute and late sequela, which have a profound impact on ultimate quality of life and return to work ability. In the present study we provide data within this unique oncologic site and demonstrate rehabilitation to be statistically correlated with positive outcomes.

Steen et al. found cervical cancer patients to be associated with 19% chronic fatigue after surgery, and even higher rates of up to 28% after the addition of chemoradiation [16]. Recent advances in the field of surgery including minimally invasive laparoscopic interventions have demonstrated improvements in perioperative outcomes specifically for endometrial cancer leading to improvements in RTW rates [17,18]. In our study, patients who received intensified RT treatment or the addition of chemotherapy experienced higher levels of symptoms. Individualized and tailored patient care is required, as more advanced tumor stages require the implementation of multimodal or more aggressive treatment, as well as enhanced support. One method to improve radiotherapy characteristics is the utilization of more advanced x-ray-based techniques including intensity modulated radiation therapy which have been shown in a prospective manner to improve side effect profiles and gynecological malignancies.

In addition to objective clinical oncologic outcomes, cancer rehabilitation aims at the preservation and restoring of normal functions among cancer survivors, as acute and long-term side effects can diminish quality of life. A new individual risk assessment is needed for identifying patients that are at a higher risk of post-therapy burdens to provide specialized supportive measures and interventions and patient information at an early stage. Fatigue and pain are wide-ranging, multi-causal symptoms, but also show multifactorial interplay with each other and often accompany depression, cognitive impairment, fear, and decreased quality of life [19,20]. As a result, proper identification of risk factors and a global assessment of the etiology of fatigue and pain are critical to ultimately improve patient’s long-term outcomes with respect to quality of life and return to normal work functioning.

The participation in inpatient cancer rehabilitation therapy, generally including exercise and nutrition counseling, has led to a significant improvement of fatigue symptoms, independent of working status in our study. Supervised exercise can reduce cancer-related fatigue in women with breast cancer, but not for prostate cancer patients or those with hematological malignancies, even though data remain controversial and are mostly validated in the breast cancer patient population [21,22,23]. Even if conclusions can only be drawn cautiously, as the nature of therapeutic interventions and patient compliance of the women in our cohort were retrospectively analyzed, our results indicate that exercise could also be beneficial for primary pelvic gynecologic malignancies.

Careful attention should be given to the physical and psychological impact of cancer treatment on younger women, as in our cohort this group of patients (<45 years) showed significantly increased risks for acute urinary toxicity and fatigue symptoms during RT as well as depression in follow up. The latter represents a serious consequence that has been described previously for young cancer patients [24,25]. Knowledge about this is important to integrate and initiate targeted efforts to inform and support patients. Furthermore, screening among cancer patients is important, as they may face not only psychological but also social, physical, or religious stresses.

Specialized assistance and support services are suitable for keeping patients connected to their professional lives, which may have other benefits aside from economic factors. In the gradual return to normalcy following cancer treatment, inability to RTW can have financial burdens as well as social consequences. The time frame for RTW of our working-age cohort among the progression-free patients was 58.1% at 12 months and 47.1% in the overall group after curative pelvic primary RT treatment, and highly depended on treatment intensity, the presence of fatigue and pain, and the implementation of inpatient post-treatment cancer rehabilitation.

Data for RTW for only pelvic gynecological cancer are rather sparse and mainly focus on cervical cancer patients. Nakamura et al. studied 97 patients with cervical cancer and compared the influence of multimodal treatment concepts of surgery and radio(chemo)therapy on RTW [26]. Patients with upfront surgery and lymphedema showed significantly lower RTW rates. Contrary results were found by Sun et al., who found that cervical cancer patients were more likely to RTW after surgical than RT treatment, but also considered vocational influencing factors, such as company size and monthly income, as highly predictive elements [27]. Saim et al. included cervical cancer patients as a subgroup of patients and reported comorbidities and fatigue leading to a higher likelihood of job loss [28]. Finally, Mamguem Kamga et al. presented long-term occupational outcomes for 66 employed women with pelvic gynecologic malignancies, of which 48.9% were still employed six years after their diagnosis [6]. For this, fatigue was also found to be an important predictor of RTW, but applied surgical or RT treatment modalities were not assessed. Due to the wide-ranging and partly controversial data, more knowledge is necessary to determine the best way for interdisciplinary teams to assist in reintegrating the patient into the workplace.

For breast cancer, however, non-RTW rates for recurrence-free women at 24 months were lower (16% [11] vs. 36.8% at 18 months in our series), and were also associated with a negative impact of intensified tumor treatment with chemotherapy and extended surgery [29]. In breast cancer, RTW rates also correlated with the type of work and job demands, such as heavy lifting, which was reported to be associated to RTW at 12 months [14,29]. While proper bladder and bowel movements certainly affect quality of life and daily activities, the overall RTW numbers following treatment for pelvic malignancies appear to be lower compared to other cancer sites. This may implicate the more massive impacts of deteriorated gastrointestinal and genitourinary functions on normal life than the range of motion of upper extremities following breast cancer treatment.

Caumette et al. reported inferior rates of RTW related to household details, when living with a partner, especially for women aged over 50 years and for women with children [9]. In our study, self-employed women were more successful from a RTW standpoint, however, the overall financial burden of cancer diagnosis and treatment in contrast to salaried women was retrospectively not assessed.

Despite low-threshold access in our cohort, only 56.8% of the women consulted specialized in-house healthcare workers during RT. Prior analyses of underutilization or barriers in access have reported that male patients and an age of older than 84 years were significantly associated with lower rates of social service utilization [7]. However, utilization might not always reflect or measure the quality of a consultation or cancer center, but may be optimized by adjusting the structure of an oncology department or cancer center to improve patient access [7]. Comprehensive assessment of health and social impairment by social services is time consuming within clinical routine, but these numbers indicate that the organization and implementation of structured counseling can maintain a high supply of service. As social services represent a strategic role, major emphasis of the need for implementation are needed. Moreover, as patients’ requirements for psychological, physical, and social needs change during the course of treatment, assessments need to be surveyed and adapted accordingly.

Our study represents and describes patient characteristics, treatment data, results of RTW, and social systems for women experiencing cancer treatment in a high-income country. Of note, lifestyle factors and access to supporting interventions underly international differences, and cannot be simply transferred. Even in developed countries, mismatches in time-consuming individualized care and the need for intensified cancer rehabilitation delivery are challenging, but these affect low- or middle-income-country patients on a different scale in terms of financial or social backgrounds as well as organization, access to, and implementation of cancer rehabilitation programs [30,31].

The primary limitation of the present study is its retrospective analysis. As a consequence, employment status was difficult to differentiate between full and part-time. In addition, details of the physical requirements, workplace environment, and employer support were not robustly analyzed for their association with RTW. Furthermore, no standardized quality of life questionnaires could be used to measure the extent of occurring symptoms to reflect each individual’s feelings of impairment. Of note, the statistical results and subgroup analyses in our study must be interpreted cautiously for potential multiplicity as a result of multiple testing and censoring in RTW rates. Nonetheless, our study presents new data for RTW in the context of cervical as well as vaginal, vulvar, and endometrial cancers for a large German cohort, focusing on curative RT treatment and, thus, provides new evidence for recovery from cancer therapy for primary pelvic malignancies.

Further systematic prospective research is needed to identify the specific reasons for unsuccessful vocational reintegration, considering not only patients’ burdens, but particularly influencing working environment factors that may hinder full vocational capacity. Moreover, detailed future insight on the extent of resulting financial difficulties needs to be assessed. Further, prospective randomized controlled trials are needed to evaluate the most effective cancer rehabilitation procedures and interventions and to provide more knowledge on the impact of physical or nutritional training on the rehabilitation of women with pelvic gynecologic malignancies. More knowledge is required and more focus needs to be put on the efficiency and quality in the delivery of cancer rehabilitation services to examine the best practices for the prevention of cancer sequelae, for efficient integration of psychosocial care, and to ensure appropriate supportive measures.

## 5. Conclusions

In the present study, we identified fatigue and pain as critical components contributing to return to work barriers following curative treatment for pelvic malignancies. Individualized inpatient cancer rehabilitation treatment resulted in high rates of returning to work after curative RT. Pivotal focus needs particularly to be placed on younger patients who showed higher appearance of acute fatigue and urinary toxicity during RT and, moreover, higher incidences of treatment-requiring depression during follow-up.

## Figures and Tables

**Figure 1 cancers-14-02330-f001:**
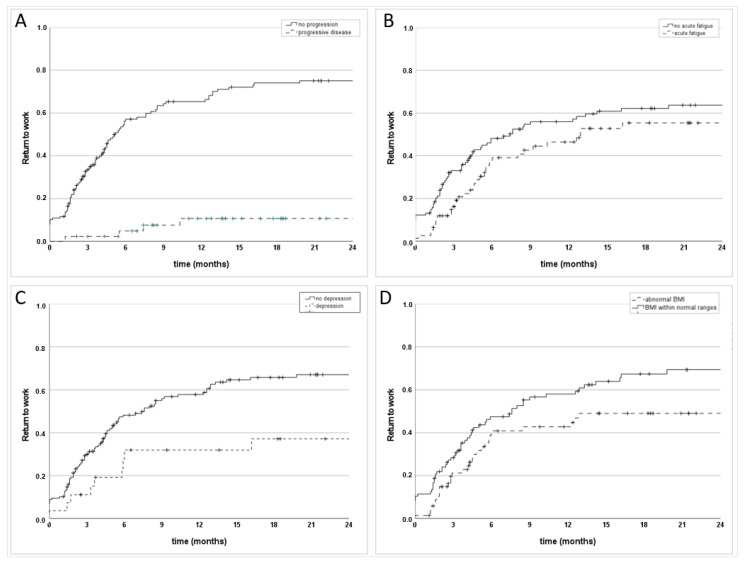
Return to work rates for the overall group in Kaplan-Meier estimates (one minus survival) depending on oncologic progression (**A**), the presence of acute fatigue during radiotherapy (**B**), the presence of depression (**C**), and body mass index (BMI) (**D**).

**Table 1 cancers-14-02330-t001:** Patient and treatment characteristics.

Characteristics	Values (Range or Percentage)
**Median age (years)**	61 (28–95)
**Age (years)**	
≤30	3 (0.7%)
31–40	30 (7.1%)
41–50	56 (13.2%)
51–60	120 (28.3%)
61–70	115 (27.1%)
71–80	73 (17.2%)
81–90	25 (5.9%)
91–100	2 (0.5%)
>100	0 (0%)
**FIGO stage**	
1	171 (40.3%)
2	77 (18.2%)
3	150 (35.4%)
4	26 (6.1%)
**Median Body Mass Index prior to RT (kg/m^2^)**	25.2 (15.6–61.7)
**Median Karnofsky performance score prior to RT (%)**	90 (40–100)
**Median adjusted * Charlson comorbidity index prior to RT**	2 (0–9)
**Year of RT treatment**	
2017	104 (24.5%)
2018	79 (18.6%)
2019	82 (19.3%)
2020	75 (17.7%)
2021	84 (19.8%)
**Median treatment time (days)**	42 (1–103)
**RT technique**	
IMRT + brachytherapy	284 (67.0%)
Brachytherapy only	106 (25.0%)
IMRT only	30 (7.1%)
**Boost**	
Simultaneously integrated	42 (9.9%)
Brachytherapy	284 (67.0%)
**Prior surgical lymph node dissection**	
Yes	301 (71.0%)
No	123 (29.0%)
**Extended field radiation**	
Including para-aortic region	47 (11.1%)
No extended para-aortic field	377 (88.9%)

IMRT: intensity-modulated radiotherapy, FIGO: International Federation of Obstetrics and Gynecology, RT: radiotherapy; * the Charlson comorbidity index [15] was utilized with an adjusted scoring for the category of “solid tumor”: points were counted only if a second malignancy, other than the treated one, was present.

**Table 2 cancers-14-02330-t002:** Return to work rates.

Group	3 Months	6 Months	9 Months	12 Months	18 Months
**(a) Overall group (*n* = 192)**	26.2%	41.4%	46.1%	47.1%	51.3%
**Primary tumor**					
Uterine cancer (*n* = 65)	35.4%	47.7%	52.3%	52.3%	55.4%
Cervical cancer (*n* = 101)	21.8%	39.6%	45.5%	47.5%	52.5%
Vulvar/vaginal cancer (*n* = 26)	19.2%	30.8%	30.8%	30.8%	34.6%
**(b) Progression-free group (*n* = 155)**	32.3%	51.0%	56.8%	58.1%	63.2%
**Primary tumor**					
Uterine cancer (*n* = 62)	37.1%	50.0%	54.8%	54.8%	58.1%
Cervical cancer (*n* = 78)	27.8%	50.6%	58.2%	60.8%	67.1%
Vulvar/vaginal cancer (*n* = 15)	33.3%	53.3%	53.3%	53.3%	60.0%

## Data Availability

The data presented in this study was obtained from local databases of the Cancer Registry of the National Center for Tumor Diseases (NCT). The data are not publicly available due to Local Ethics Committee rules.

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
