# Peer review of "Return to Work, Fatigue and Cancer Rehabilitation after Curative Radiotherapy and Radiochemotherapy for Pelvic Gynecologic Cancer"

_cancers, 2022, doi:10.3390/cancers14092330_

Round 1
Reviewer 1 Report
The authors conducted a retrospective analysis of data from patients with primary pelvic gynecologic malignancies treated with curative radiation therapy or chemoradiation at a single institution between January 2017 and December 2021. They aimed to identify factors associated with cancer rehabilitation and returning to work. The below comments primarily focus on clarifying the analyses, including the authors’ assumptions regarding missing data as well as the number and type of significance tests conducted.
It is unclear how much data were available for these patients. For example, the authors state that “Current working status was assessed at baseline and during follow-up at different timepoints” (p. 4). Given the wide range of follow-up times, what was the minimum/median/maximum number of times that current working status was assessed in these patients? And what censoring rules were used to generate Figures 1A to 1D? The authors should not assume that current working status remains negative indefinitely after the last time a patient’s current working status was assessed.
It is unclear how the authors dealt with missing data. For example, in the Discussion section (also see the abstract and Table 2), the authors highlight that “only 58.1% of progression-free women returned to work at 12 months” (p. 9). However, the median follow-up time was only 11.6 months in the entire sample (and not reported for the subsample of progression-free working patients). As such, employment status would be unknown for a large proportion of the sample. Please clarify.
“Employment status was assessable for a subgroup of 142 working patients: 133 (93.7%) were salaried and 9 (6.3%) were self-employed” (p. 7). What criteria were used to determine whether employment status was assessable? The Methods section should state those criteria.
Immediately after stating that employment status was only assessable in 142 working patients, the authors state the following. “During the follow-up period, 101 (52.9%) women returned to work after a median time of 3.2 (range: 0–33.1) months, while 91 (47.6%) patients did not return” (p. 7). However, 101+91=192, not 142. It is unclear why this analysis and subsequent analyses (e.g., those presented in Figure 1 and Table 2) would include working patients whose employment status was not assessable.
What was the minimum/median/maximum follow-up period for working patients? The follow-up period was as low as 0.2 months in the full sample.
Throughout the Results section, the authors frequently report p-values in isolation without reporting proportions, means, confidence intervals, etc. that would show the magnitude of the differences or associations. Providing this information would help convey the clinical significance of the results.
In the Results section, it is difficult to discern which variables are continuous and which variables are categorical (and, if so, which categories were used).
“Baseline” is not clearly defined on page 4. Does “baseline” refer to the start of radiation therapy? The date of diagnosis? Did it differ across patients?
The authors conducted a very large number of significance tests, leading to concerns about multiplicity. At minimum, this limitation should be addressed in the Discussion section.
Related to the above comment, more information is needed in Section 2.4 (“Statistical Analysis”). The full list of variables tested for significance is unclear.
It is unclear which age categories were used during analysis. Table 1 uses 10-year increments. Page 6 states that “Urinary toxicity was…more significant among women aged less than 40 years (p=0.036).” Was age dichotomized as less than 40 years old versus 40 years old or older? The abstract and page 6 state that “Women aged less than 45 years at the start of RT were significantly more in danger of developing depression (p=0.001) after the end of treatment than older women.” Was age dichotomized as less than 45 years old versus 45 years old or older? Using different dichotomizations/categorizations again raises concerns about multiplicity.
Minor Comments
“The Charlson-Comorbidity-Index reliably predicted patients associated with significantly higher rates of fatigue” (p. 1). Please reword this statement for clarity.
In Table 1, please include a footnote describing the alternative scoring used for the Charlson comorbidity index.
Line 265 states that 191 patients were working full- or part-time, but other analyses use N=192 (e.g., Table 2, results presented on p. 7 and p. 9).
“Nausea, gastrointestinal toxicity, and dermatitis/mucositis were more present in women with higher FIGO stages and intensified treatments (chemotherapy (p<0.001), more fractions (p<0.001), and higher RT doses (p<0.001))” (p. 6). Did the authors create a composite of nausea, gastrointestinal toxicity, and dermatitis/mucositis? If not, I would expect separate p-values for nausea, gastrointestinal toxicity, and dermatitis/mucositis.
In Section 3.3, the authors seem to switch between considering patients whose fatigue appeared during/after radiation therapy and patients with fatigue at any time (including before radiation therapy). Please clarify which groups are being compared.
Author Response
The authors conducted a retrospective analysis of data from patients with primary pelvic gynecologic malignancies treated with curative radiation therapy or chemoradiation at a single institution between January 2017 and December 2021. They aimed to identify factors associated with cancer rehabilitation and returning to work.
The below comments primarily focus on clarifying the analyses, including the authors’ assumptions regarding missing data as well as the number and type of significance tests conducted. It is unclear how much data were available for these patients. For example, the authors state that “Current working status was assessed at baseline and during follow-up at different timepoints” (p. 4). Given the wide range of follow-up times, what was the minimum/median/maximum number of times that current working status was assessed in these patients? And what censoring rules were used to generate Figures 1A to 1D? The authors should not assume that current working status remains negative indefinitely after the last time a patient’s current working status was assessed.
>>>>> Thank you for very much for your comments and helpful suggestions, we do totally agree that further detailed data regarding the assessment of the working status is required. To avoid any misunderstanding, we added more information and revised the corresponding paragraph on how the current working status was assessed and defined as baseline status in the “Methods” section and how censoring was performed. All changes can be viewed by “track changing function” in Word.
We added more detailed data on how a “successful” return to work status (RTW) was defined and documented during follow-up appointments as well as follow-up intervals. Recommended revisit follow-up intervals were every three months during the first and second year after treatment, every 6 months in the third year and once a year after that for at least five years. Furthermore, we inserted the subgroup minimum/median/maximum follow-up times for the working subgroup with a median follow-up of 20.8 (range: 1.2 – 62.4) months.
Unfortunately, due to the retrospective nature of the study, only the first time point of vocational reintegration was reliably assessable. 16 patients (8.3%) within the subgroup of working women were lost-to-follow-up after 15 months, for the remaining 176 women (91.7%) full follow-up data of at least 18 months were assessable. Return to work status was assessed during treatment and follow-up by reviewing inhouse or referring physician’s notes, inpatient cancer rehabilitation discharge letters and social worker’s databases and documented as successful at the time point of the first reintegration into employment. The figures present Kaplan-Meier (one minus) estimates, where women who had not returned to work before the last contact were censored.
As we already stated in the “Discussion” section, we are aware of these retrospective limitations and that there is a need for further prospective evaluation of influencing factors concerning workplace environment, dynamic changes throughout follow-up, employer support or full- and part-time employment and specific reasons for non-reintegration. Nevertheless, retrospective data for pelvic malignancies are sparse regarding cancer rehabilitation and RTW. To account for and emphasize this, we additionally expanded the limitations/future research section in the Discussion.
It is unclear how the authors dealt with missing data. For example, in the Discussion section (also see the abstract and Table 2), the authors highlight that “only 58.1% of progression-free women returned to work at 12 months” (p. 9). However, the median follow-up time was only 11.6 months in the entire sample (and not reported for the subsample of progression-free working patients). As such, employment status would be unknown for a large proportion of the sample. Please clarify.
>>>>> To clarify and avoid any misunderstanding of our data and results, we added the subgroup minimum/median/maximum follow-up times for the working subgroup with a median follow-up of 20.8 (range: 1.2 – 62.4) months, which was longer than for the overall cohort (median 11.6 (range: 0.2 – 65.6) months). 16 patients (8.3%) within the subgroup of working women were lost-to-follow-up after 15 months, for the remaining 176 women (91.7%) full follow-up data of at least 18 months were assessable. For this reason, we only provided return to work rates for 3, 12 and 18 months in the abstract and added data about lost-to-follow-up patients. To further avoid any misinterpretation or any inaccurate estimation, we removed the results of the percentages for 24 months in the manuscript, that were listed in Table 2 and the discussion section.
“Employment status was assessable for a subgroup of 142 working patients: 133 (93.7%) were salaried and 9 (6.3%) were self-employed” (p. 7). What criteria were used to determine whether employment status was assessable? The Methods section should state those criteria.
>>>>> We corrected and specified our English wording concerning “active working status” (n=191 at baseline) and “employment status”, which might have led to misunderstandings. Within the group of working women (n=191 at baseline), there were additionally assessable data for 142 patients to further describe their employment characteristics, which we were only referring to regarding “salaried- or self-employment status”. We added the limited data for this subgroup in the corresponding results.
Immediately after stating that employment status was only assessable in 142 working patients, the authors state the following. “During the follow-up period, 101 (52.9%) women returned to work after a median time of 3.2 (range: 0–33.1) months, while 91 (47.6%) patients did not return” (p. 7). However, 101+91=192, not 142. It is unclear why this analysis and subsequent analyses (e.g., those presented in Figure 1 and Table 2) would include working patients whose employment status was not assessable.
>>>>> As we already mentioned above, there seems to be a misunderstanding, caused by the incorrect use of the word “employment status”, which we have corrected throughout the manuscript.
Data were assessable for a median follow-up time of 20.8 months for the entire subgroup of 191 working patients, of which 101 patients had successfully returned to work, while 91 did not. 101 + 91 =192, which led to a mismatch of 1 woman, that was working age and counted as unemployed in baseline measure (as she had lost her job only a few days prior to the start of definitive RT at diagnosis), but was able to return to employment only a few days after the end of radiotherapy.
What was the minimum/median/maximum follow-up period for working patients? The follow-up period was as low as 0.2 months in the full sample.
>>>>> As clarified above, we have added the subgroup minimum/median/maximum follow-up times (median 20.8 (range: 1.2 – 62.4) months for the working subgroup, which was longer than for the overall cohort (median 11.6 (range: 0.2 – 65.6) months).
Throughout the Results section, the authors frequently report p-values in isolation without reporting proportions, means, confidence intervals, etc. that would show the magnitude of the differences or associations. Providing this information would help convey the clinical significance of the results. In the Results section, it is difficult to discern which variables are continuous and which variables are categorical (and, if so, which categories were used).
>>>>> We added more information about our statistical analyses in the “Methods” and the “Results” sections for a better identification of the levels of measurements and continuous and categorial variables used, including median, quartiles, hazard rations and 95%-confidence intervals. We kindly ask to submit the detailed statistical results including means, quartiles and proportions as supplementary material (Table 2 and 3) to simply reading.
“Baseline” is not clearly defined on page 4. Does “baseline” refer to the start of radiation therapy? The date of diagnosis? Did it differ across patients?
>>>>> We have added definitions of the term “baseline” in the “Methods” section. To account for different treatment concepts with upfront surgery or definitive radiotherapy, the definition of baseline for the current working status was defined at the time of diagnosis. For the evaluation of toxicity and symptoms, baseline was defined prior to the start of RT as assessed in clinical routine at the first outpatient appointment at our radio-oncologic department.
The authors conducted a very large number of significance tests, leading to concerns about multiplicity. At minimum, this limitation should be addressed in the Discussion section. Related to the above comment, more information is needed in Section 2.4 (“Statistical Analysis”). The full list of variables tested for significance is unclear.
>>>>> We added more information about our statistical analyses in the “Methods” and the “Results” sections for a better identification of the levels of measurements and continuous and categorial variables used, including median, quartiles, hazard rations and 95%-confidence intervals. We have further added this concern to the discussion / limitations section, to emphasize the need for a critical interpretation of our retrospective results. We kindly ask to submit the detailed statistical results including means, quartiles and proportions as supplementary material (Table 2 and 3) to simply reading.
It is unclear which age categories were used during analysis. Table 1 uses 10-year increments. Page 6 states that “Urinary toxicity was…more significant among women aged less than 40 years (p=0.036).” Was age dichotomized as less than 40 years old versus 40 years old or older? The abstract and page 6 state that “Women aged less than 45 years at the start of RT were significantly more in danger of developing depression (p=0.001) after the end of treatment than older women.” Was age dichotomized as less than 45 years old versus 45 years old or older? Using different dichotomizations/categorizations again raises concerns about multiplicity.
>>>>> Patient characteristics included age intervals (10-year intervals: ≤ 30, 31-40, 41-50, 51-60, 61-70, 71-80, 81-90, 91-100 years) that were listed in Table 1 for a better presentation of our patient cohort. We added a line for women >100 years (n=0, 0%) in Table 1 in the re-submitted manuscript. For statistical analysis, however, receiver operating characteristic (ROC) curve analysis was performed to calculate a cut-off age value for depression and age was then dichotomized (<45 vs. ≥45 years). For a better comparison and to avoid multiple testing of different age intervals we re-assessed the tests for the same age (<45 vs. ≥45 years), including urinary toxicity (remained p=0.036) and changed it in the manuscript. Moreover, we do agree that statistical and subgroup analyses must be interpreted cautiously and additionally added a section for this to emphasize the need for a critical evaluation of our results and potential multiplicity in the limitations in the “Discussion”.
Minor Comments
“The Charlson-Comorbidity-Index reliably predicted patients associated with significantly higher rates of fatigue” (p. 1). Please reword this statement for clarity.
>>>>> We have reworded the sentence: “A higher Charlson-Comorbidity-Index reliably predicted patients associated with a higher risk of acute fatigue during RT”
In Table 1, please include a footnote describing the alternative scoring used for the Charlson comorbidity index.
>>>>> We have included a footnote to provide the additional information about the adjustment of the Charlson Comorbidity Index.
Line 265 states that 191 patients were working full- or part-time, but other analyses use N=192 (e.g., Table 2, results presented on p. 7 and p. 9).
>>>>> As we already mentioned above, there were 191 working patients at baseline, of which 101 patients had successfully returned to work, while 91 did not. 101 + 91 =192, which led to a mismatch of 1 woman, that was working age and counted as unemployed in baseline measure (as she had lost her job only a few days prior to the start of definitive RT at diagnosis), but was able to return to employment only a few days after the end of radiotherapy. We have clarified this mismatch in the manuscript to avoid any misunderstanding.
“Nausea, gastrointestinal toxicity, and dermatitis/mucositis were more present in women with higher FIGO stages and intensified treatments (chemotherapy (p<0.001), more fractions (p<0.001), and higher RT doses (p<0.001))” (p. 6). Did the authors create a composite of nausea, gastrointestinal toxicity, and dermatitis/mucositis? If not, I would expect separate p-values for nausea, gastrointestinal toxicity, and dermatitis/mucositis.
>>>>> We did not create a composite, each category of toxicity was considered independently of each other, but p-values were summarized with the same statistical p-value of p<0.001 to simplify reading. To clarify this and give more detailed information about the statistical results, we adapted the paragraph and presented the statistical results for each category.
In Section 3.3, the authors seem to switch between considering patients whose fatigue appeared during/after radiation therapy and patients with fatigue at any time (including before radiation therapy). Please clarify which groups are being compared.
>>>>> The presence and CTCAE grading of fatigue symptoms were assessed at baseline, during RT (presence and highest CTCAE grading) and during follow-up (presence and highest CTCAE grading). We specified this in the re-submitted manuscript in the “Methods” and “Results” section to clarify what we were referring to.

Reviewer 2 Report
Abstract - Reads well
Introduction - Reads well
Methods
Good explanation but more details required about how data was collected?
How were details about depression collected?
How were details about fatigue collected?
Ethics –
Were participants informed about the study? Did participants consent for this data to be used in this analysis? Did participants consent to having their smoking status and BMI reported?
Results
Demographics
Put household details prior to smoking sentences
What nationality were participants?
Details about different components
Please provide details about in-house social service consultations and in house counselling
What is specialised inpatient cancer rehabilitation
How was return to work data collected from participants?
Discussion well written
I would like to see more consideration of interventions to support women and future research needed
conclusion
does the last sentence in the conclusion reflect the findings - perhaps more of a focus on younger people could be made in other sections of the manuscript if this is the case.
Author Response
Abstract - Reads well. Introduction - Reads well.
Methods: Good explanation but more details required about how data was collected? How were details about depression collected? How were details about fatigue collected?
>>>>> Thank you for very much for your comments and helpful suggestions. We added more information to the “Methods” section concerning the revisit follow-up intervals and the assessment of acute and late toxicity as well as depression and fatigue. The presence and CTCAE grading of fatigue symptoms were assessed at baseline, during RT (presence and highest CTCAE grading) and during follow-up (presence and highest CTCAE grading). We specified this in the re-submitted manuscript in the “Methods” and “Results” section to clarify what we were referring to. All changes in the re-submitted manuscript can be viewed by “track changing function” in Word.
The presence of depressive symptoms was documented by physicians during routine visits at baseline and follow-ups and included the requirements of medical or psychological treatment. Of note, due to the retrospective nature of the study, depression scale questionnaires could not be applied, but physician’s documentation of the presence and ratings of depressive symptoms and treatment requirements were assessed. Acute and late toxicities were assessed according to the Common Terminology Criteria for Adverse Events (CTCAE, version 5.0).
Ethics – Were participants informed about the study? Did participants consent for this data to be used in this analysis? Did participants consent to having their smoking status and BMI reported?
>>>>> The study was conducted according to the guidelines of the Declaration of Helsinki, and approved by the local ethics committee of the University Hospital of Heidelberg (S-453/2021) before the review of data. Due to this approval and local regulatory requirements, participants could be included into this retrospective analysis without being informed and without giving each individual’s written informed consent. This also included the smoking status and BMI. The manuscript does not contain any identifiable material or content, all data were anonymized. A copy of the ethic approval was provided to the journal at the time of submission.
Results: Demographics, Put household details prior to smoking sentences. What nationality were participants?
>>>>> We have changed the order for household details and smoking status and included the distribution of nationalities. The population was predominantly White/Caucasian (n=401, 94.6 %) and only minimal representation of North Asians (n=13, 3.1%), South Asians (n=3, 0.7%), Sub-Saharan (n=5, 1.2%), and other (n=2, 0.4%). Thus, subgroup analyses were not considered to be reliable and were not performed for ethnicity groups.
Details about different components: Please provide details about in-house social service consultations and in house counselling. What is specialised inpatient cancer rehabilitation
>>>>> We have provided more detailed information on what tasks were undertaken and initiated by the interdisciplinary team during in-house social service consultations in the “Methods” section. Further, we defined specialized inpatient cancer rehabilitation and its requirements and procedures.
How was return to work data collected from participants?
>>>>> We added more data and revised the corresponding paragraph on how the current working status was assessed and defined at baseline in the “Methods” section. Further, we inserted more detailed data on how a “successful” return to work status was defined and assessed during follow-up. Unfortunately, due to the retrospective nature of the study, only the first time point of vocational reintegration was reliably assessable. As already stated in the “Discussion” section, we are aware of these limitations and that there is a need for further prospective evaluation of influencing factors concerning workplace environment, employer support or full- and part-time employment. Nevertheless, retrospective data for pelvic malignancies are sparse regarding cancer rehabilitation and RTW. To account for and emphasize this, we additionally expanded the limitations/future research section in the Discussion.
Discussion well written. I would like to see more consideration of interventions to support women and future research needed
>>>>> Regarding the above-mentioned limitations, we extended the “Discussion” and particularly the limitations/future research section to emphasize the need for more clinical prospective research concerning the specific reasons for unsuccessful vocational reintegration. Moreover, detailed future insight needs to be put on the extent of resulting financial difficulties. Further, prospective randomized controlled trials are needed to evaluate the most effective cancer rehabilitation procedures and interventions and to provide more knowledge on the impact of physical training on the rehabilitation of women with pelvic gynecologic malignancies.
Conclusion: does the last sentence in the conclusion reflect the findings - perhaps more of a focus on younger people could be made in other sections of the manuscript if this is the case.
>>>>> Women below 45 years suffered significantly more often from acute urinary toxicity, acute fatigue symptoms and the development of therapy-requiring depression during follow-up. For a better comparison and to avoid multiple testing of different age intervals we re-assessed the variables and statisticals tests for the same age (<45 vs. ≥45 years), including urinary toxicity and fatigue during RT. Moreover, we changed the corresponding paragraphs in the manuscript and conclusion: “Pivotal focus needs particularly to be placed on younger patients who showed higher appearance of acute fatigue and urinary toxicity during RT and, moreover, higher incidences of treatment-requiring depression during follow-up.”
